# Long-Term Outcomes of Laparoscopic Greater Curvature Plication and Laparoscopic Sleeve Gastrectomy: Critical Appraisal of the Role of Gastric Plication in Bariatric Surgery

**DOI:** 10.3390/medicina58091299

**Published:** 2022-09-17

**Authors:** Lien-Cheng Tsao, Joseph Lin, Bing-Yen Wang, Yu-Jun Chang, Cheng-Yen Huang, Shu-Fen Yu, Wan-Yu Hung, Chi-Chien Lin, Chien-Pin Chan

**Affiliations:** 1Department of General Surgery, Changhua Christian Hospital, Changhua 500, Taiwan; 2Institute of Biomedical Science, College of Life Sciences, National Chung-Hsing University, Taichung 402, Taiwan; 3Department of Thoracic Surgery, Changhua Christian Hospital, Changhua 500, Taiwan; 4Epidemiology and Biostatistics Center, Changhua Christian Hospital, Changhua 500, Taiwan; 5Department of Nursing, Changhua Christian Hospital, Changhua 500, Taiwan

**Keywords:** laparoscopic sleeve gastrectomy, laparoscopic greater curvature plication, comorbidity, reoperation, weight reduction

## Abstract

*Background and Objectives*: This single-center study aimed to assess the role of laparoscopic greater curvature plication (LGCP) in bariatric surgery. *Materials and Methods*: Using data from our institution’s prospectively maintained database, we identified adult patients with obesity who underwent either laparoscopic sleeve gastrectomy (LSG) or LGCP between January 2012 and July 2017. In total, 280 patients were enrolled in this study. *Results*: The body mass index was higher in the LSG group than in the LGCP group (39.3 vs. 33.3, *p* < 0.001). Both groups achieved significant weight loss during the 3-year follow-up (*p* < 0.001). The weight-reduction rate was higher in the LSG group than in the LGCP group 6, 12, and 24 months postoperatively (*p* = 0.001, 0.001, and 0.012, respectively). The reoperation rate of the LGCP group was higher than that of the LSG group (*p* = 0.001). No deaths were recorded in either group. *Conclusions*: Although both the LGCP and LSG groups achieved significant weight loss over three years, the LGCP group demonstrated a lower weight-reduction rate and a higher reoperation rate than the LSG group. Thus, it is necessary to reassess the role of LGCP in bariatric surgery, particularly when LSG is a feasible alternative.

## 1. Introduction

Obesity has become a severe public health issue worldwide. In 1999, studies revealed that more than half of the adults in the USA were overweight or obese. Obesity was also identified as a major cause of mortality, with 28,000 adult deaths per year in the USA [1,2]. Twenty years later, the World Health Organization (WHO) announced that at least 2.8 million people globally were dying annually because of overweight or obesity, not only in high-income but also in low- and middle-income countries [3]. Furthermore, in 2020, researchers found that patients with obesity have a high mortality risk when contracting coronavirus disease 2019 (COVID-19) [4,5]. Therefore, weight reduction has become a critical health strategy in patients with overweight and obesity worldwide. Bariatric surgery has become a method of choice for weight reduction in developed and developing countries if conventional measures fail. In 2003, more than 100,000 bariatric procedures were performed in the USA [6].

Laparoscopic sleeve gastrectomy (LSG) is the most commonly performed bariatric procedure; it is safe and feasible in reducing weight and improving obesity-related comorbidities [7,8]. Alternatively, laparoscopic greater curvature plication (LGCP) is a new procedure considered to be easy to perform and reversible that has a lower complication rate than LSG. Short-term weight reduction and improvement of comorbidities after LGCP have been described, and it does not remove parts of the stomach [9,10,11,12]. Both LSG and LGCP employ volume restriction of the stomach to achieve weight loss. Previous studies comparing the outcomes of these two procedures on weight reduction, perioperative complications, and improvement of comorbidities have revealed that LSG led to higher weight-reduction levels than LGCP [13,14,15,16,17,18,19,20,21,22,23]. Another case-matched control study demonstrated that both LGCP and LSG are safe and feasible in surgical weight reduction; however, LSG achieved more significant excess body weight reduction in the short term [24].

Herein, we report a 3-year follow-up study to define the long-term outcomes of LGCP and LSG performed in our medical center. The results indicated that LGCP demonstrated a lower weight-reduction rate but a higher reoperation rate than LSG, even though both bariatric procedures achieved significant weight loss. Accordingly, we argue that the role of LGCP in bariatric surgery should be re-evaluated, especially when LSG is a feasible alternative.

## 2. Materials and Methods

This study was designed as a retrospective file review. The authorization of the institutional review board (IRB) of Changhua Christian Hospital was obtained (IRB No. 200807) before commencing data collection. All procedures performed in this study followed the ethical standards of the IRB. Informed consent was waived because data were collected retrospectively.

### 2.1. Patients

A total of 387 patients who underwent LSG and LGCP procedures for morbid obesity at Changhua Christian Hospital between January 2012 and July 2017 were included. Subsequently, 107 patients were excluded from this study based on the following criteria: (a) other bariatric surgeries for morbid obesity, (b) first bariatric surgery performed at other hospitals, and (c) lost to follow-up below 3 years. The indication for surgery was in accordance with the suggestions for preventing and managing morbid obesity in Asian populations by the WHO, namely, body mass index (BMI) ≥ 35 kg/m^2^ with or without comorbidities and BMI ≥ 27.5 kg/m^2^ with medically treated comorbidities [25]. All patients took part in our shared decision-making process and received detailed information on the advantages and disadvantages of LSG and LGCP. The procedures were selected considering each patient’s anatomy. In this study, we distinguished two groups of patients as the LGCP and LSG groups.

### 2.2. Variables

We recorded patients’ demographic and clinical data, including sex, age, baseline BMI, weight excess, operative time, duration of hospital stay, comorbidities, and relevant biochemical data. Follow up was conducted in compliance with the current guidelines: 2 and 6 weeks postoperatively and every 3 months during the first year. It was followed by every 6 months during the second year and annually after that. 

We defined perioperative morbidity and mortality rates as the primary endpoints. The secondary endpoints were operative time, length of hospital stay, BMI, percentage of excess weight loss (% EWL), percentage of total weight loss (% TWL), estimated weight-reduction rate, reoperation, complications (such as acute respiratory failure, pneumonia, electrolyte imbalance, pulmonary edema, intraabdominal abscess, gastric outlet obstruction, etc.), and improvement of comorbidities. A diagnosis of fatty liver was made based on the results of abdominal ultrasonography performed by trained technicians. All images and data associated with a particular scan were stored electronically; they were later reviewed by gastroenterologists to make the diagnosis of fatty liver without referencing any of the participants’ other individual data.

### 2.3. Surgical Technique

All patients were placed in a modified reverse Trendelenburg position with both arms abducted. Both elastic and intermittent pneumatic compression stockings were used. We made 3–4 skin incisions of 5 mm and 10 mm length, including the trans-umbilical incision as the camera port. Both procedures started with the dissection of the greater gastric curvature from 4 cm proximal to the pylorus to the angle of His, and pouch calibration was achieved by passing a 32-Fr endoscope toward the pylorus.

In LSG, we performed vertical resection of the greater gastric curvature using a stapling device, starting 4–5 cm proximal to the pylorus with a green cartridge and following with a series of blue cartridges. A 3-0 V-Loc™ (Medtronic, Dublin, Ireland) was applied, followed by a continuous running suture [26,27,28]. In LGCP, we applied two rows of extramucosal sutures to achieve plication. The first row of interrupted stitches was performed using 2-0 silk sutures and the second row of interrupted stitches with 2-0 Ti-Cron™ (Medtronic) sutures [24].

In both LGCP and LSG, a drain was placed parallel to the gastric pouch. Upper gastrointestinal endoscopy was routinely performed to assess the final stomach capacity intraoperatively. Oral fluids were allowed after the first flatus passage. Patients were maintained on an oral liquid diet for at least 2 weeks postoperatively before soft foods were started. Solid foods were gradually introduced thereafter.

### 2.4. Statistical Analysis

We used descriptive statistics for the demographic and other clinical data of the patients. Continuous variables were presented as the medians and interquartile ranges (IQR, 25th–75th percentile), whereas categorical variables were presented as numbers and percentages. The Mann–Whitney U test was used to compare the median values of continuous variables between the LGCP and LSG groups, whereas the chi-square test or Fisher’s exact test were used for categorical variables.

We also evaluated the changes in the clinical data of each patient over time postoperatively. To compare the weight loss effects and longitudinal changes between the two groups, we used multiple generalized linear models with a gamma distribution and log link within the generalized estimating equations (GEE) method for the weight-reduction rate changes and adjusted for a possible intercorrelation between data from the same patient. Moreover, the estimated weight-reduction rates in both groups were analyzed and adjusted for age, sex, BMI at baseline, hyperglycemia, and a fatty liver.

All statistical analyses were performed using IBM SPSS Statistics for Windows, version 22.0 (IBM Corp., Armonk, NY, USA). A *p*-value < 0.05 was considered to indicate a significant difference.

## 3. Results

During the study period, a total of 49 patients received LGCP, and 231 patients underwent LSG. All LGCPs and LSGs were performed laparoscopically without conversion to open surgery being required. All patients analyzed were followed up for 3 years postoperatively. The LSG group had a significantly higher BMI than the LGCP group (median BMI 39.3 in LSG and 33.3 in LGCP, *p* < 0.001). Furthermore, the LSG group had higher values of serum glutamic pyruvic transaminase (GPT), uric acid, preprandial blood glucose, and hemoglobin A1c (HbA1c). The median operative time in the LGCP group was 183 (IQR, 132–193) min, which was significantly longer than the 159 min in the LSG group (IQR, 155–220) (*p* < 0.001). (Table 1). Compared with the LSG group, the LGCP group had a higher proportion of female patients and higher rates of hyperglycemia and fatty liver. The reoperation rate in the LGCP group was significantly higher than that in the LSG group (14.3% vs. 1.7%, *p* = 0.001). Four patients had postoperative complications in the LSG group, whereas no complications occurred in the LGCP group. No patients died in either group (Table 2). Notably, data tracking at different follow-up assessments postoperatively presented that both LGCP and LSG groups achieved positive results regarding the decrease in body weight, BMI, serum glutamic oxaloacetic transaminase (GOT), serum GPT, cholesterol, triglyceride, creatinine, and HbA1c levels, along with the achieved estimated weight-reduction rate (Table 1 and Figure 1).

After adjusting for confounding factors, the GEE method revealed that the LSG group achieved a steady weight-reduction 3 years following surgery. The weight-reduction rate in men was 1.097 times that of women. Moreover, we noticed that the higher the BMI at the time of surgery, the lower the rate of weight reduction. Notably, results from the analysis of each method at different time points revealed that the weight-reduction rate was higher in the LSG group than in the LGCP group 6, 12, and 24 months postoperatively (*p* = 0.001, 0.001, 0.012, respectively) (Table 3). Specifically, the estimated weight-reduction rate demonstrated a steady weight loss during the first year postoperatively in both groups. Even though the LSG group demonstrated higher median weight, BMI, glutamic pyruvic transaminase (GPT), uric acid, and HbA1c values than the LGCP group before surgery, the weight-reduction rate was higher in the LSG group than in the LGCP group more than 2 years postoperatively. The range of the weight-reduction rate in the LGCP group was narrower than that in the LSG group (from 2 years to 3 years postoperatively, 37.2–36 in the LGCP group and 58.8–56.6 in the LSG group) (Table 4). The %TWL measure also demonstrated similar results (Table 4). 

Over a 3-year period, 129 out of 280 patients (LSG: 111 and LGCP: 18) were compliant with recommendations for follow-up. Of these patients, the proportion having adequate weight loss (%EWL > 50% at nadir) were 68.5% (*n* = 76) and 33.3% (*n* = 6) for LSG and LGCP groups, respectively (*p* = 0.004). In contrast, the proportion of patients having inadequate weight loss (%EWL < 50% at nadir) throughout subsequent follow-ups were 31.5% (*n* = 35) and 66.7% (*n* = 12) for LSG and LGCP groups, respectively (*p* = 0.004). 

The LSG group was significantly heavier at baseline than the LGCP group (median BMI 39.3 in LSG and 33.3 in LGCP, *p* < 0.001), and the LSG group was also associated with significantly poorer glycemic control preoperatively. However, after being adjusted for confounding factors, this study further demonstrated that LSG method was capable of inducing significantly greater weight loss 3 years following surgery. 

Remission rates of metabolic comorbidities in groups of LSG and LGCP [type 2 DM: 19/22 (86.4%) and 4/5 (80%), Dyslipidemia: 36/39 (89.7%) and 7/8 (87.5%), hypertension: 19/26 (73.1%) and 4/6 (66.7%)] were not significantly different. 

## 4. Discussion

Obesity is an increasing health problem worldwide. As obesity became global, the term “globesity” was coined [29,30]. Obesity causes psychosocial issues, enhances the risk of many chronic diseases [31], and increases the mortality rate of viral infections such as the H1N1 influenza A virus [32] and COVID-19 [4,5,33]. The WHO defines a BMI of 18.5–24.9 kg/m^2^ as the normal body weight range and recommends people with BMI > 35 kg/m^2^ for medical treatment to improve their comorbidities [29]. Bariatric surgery, together with diet control and lifestyle changes, not only achieves long-term weight loss but also improves comorbidities and reduces the risk of cardiovascular events and deaths in adults with obesity [6,34,35].

LGCP and LSG are two of the most popular bariatric procedures. The beneficial effects of these procedures have been an intense issue of evaluation. Several studies comparing LSG with LGCP in patients with morbid obesity have concluded either similar or inferior weight loss after LGCP [13,14,15,16,17,18,19,20,21,22,23]. Another study demonstrated less excess body weight reduction after LGCP than after LSG; however, both procedures led to similar perioperative complications [24].

Over the last few decades, different bariatric surgeries have been employed and patients can effectively benefit from the goals of sustained weight loss, comorbidity risk reduction, treatment of metabolic disease, and quality of life improvement [36]. We are starting to learn that obesity is a chronic disease as more research is taking place in this area of study. Defining surgical outcomes has been well described as adequate or inadequate weight loss using the criteria of 50% EWL from the flagship study published by Halverson and Köehler in 1981 [37]. A good number of subsequent studies have also used this benchmark as a good bariatric weight loss result suggesting that EWL% is an accurate metric in this regard [38,39,40,41]. It was demonstrated that %EWL can be largely influenced by preoperative BMI, misrepresenting true weight-loss efficacy, not in favor of those with higher BMI. For example, Boza et al. reported that the patients with a preoperative BMI > 40 kg/m^2^ achieved significant lower %EWL in comparison with the patients with BMI < 40 kg/m^2^ (50.2% versus 72.7%) [42]. Mui et al. also showed that patients with a BMI < 35 kg/m^2^ appeared to obtain more significant weight loss from bariatric surgery compared with patients with a BMI > 35 kg/m^2^ [43]. The mean BMI of our patients was 39.2 kg/m^2^. In addition to that, there were only 41.1% (*n* = 116) and 6.1% (*n* = 17) of patients with a BMI of more than 40 kg/m^2^ and more than 50 kg/m^2^, respectively, in the current cohort. On the other hand, there have been studies that reported that %TWL is the least influenced by confounding factors, and it can be compared with behavioral and pharmacological series reported in the literature [44]. Moreover, %TWL has also been reported to be easy to calculate, comprehend and explain to patients [45]. Therefore, calculation of both %EWL and %TWL were reported in the current study as variables to measure weight-loss outcomes. 

In addition to weight loss, our study demonstrated the remission rates of obesity-associated co-morbidities such as type 2 DM, dyslipidemia, and hypertension during a 3-year follow up period. Remission rates of metabolic comorbidities in groups of LSG and LGCP [type 2 DM: 19/22 (86.4%) and 4/5 (80%); Dyslipidemia: 36/39 (89.7%) and 7/8 (87.5%); hypertension: 19/26 (73.1%) and 4/6 (66.7%)] were not significantly different. These were consistent with previous studies [46]. We did not have enough evidence to report our remission rate of obstructive sleep apnea due to the retrospective nature of this study; however, previous reports have shown that OSA remission rates fall within the range of 60–90% after receiving surgical treatment [46,47]. The resolution rate of comorbidity plays a crucial role in evaluating efficacy in bariatric surgery. In our study, improvement in type 2 diabetes, dyslipidemia, and hypertension were compared. No significant differences were observed between LSG and LGCP, and this indicates that these two procedures were effective for treating patients with obesity and hypertension, type 2 diabetes, or dyslipidemia.

Previous studies have demonstrated equal effectiveness of both LGCP and LSG in feasibility and safety for surgical weight reduction in short-term follow-ups [12,24]. Despite the existing guidelines for follow-up of patient undergoing bariatric surgery, regular follow-ups could be relatively difficult in most patients. Follow up was conducted in compliance with the current guidelines in the current study: 2 and 6 weeks postoperatively and every 3 months during the first year. It was followed by every 6 months during the second year and annually after that [11]. Over a 3-year period, 129 out of 280 patients (LSG: 111 and LGCP: 18) were compliant with recommendations for follow-up. Of these patients, surgical adequate rate (%EWL > 50%) was achieved in 82 of the followed patients (82/129, 63.6%). The proportion having adequate weight loss (%EWL > 50%) were 68.5% (*n* = 76) and 33.3% (*n* = 6) for LSG and LGCP groups, respectively (*p* = 0.004). In contrast, the proportion of patients having inadequate weight loss (%EWL < 50% at nadir) throughout subsequent follow-up were 31.5% (*n* = 35) and 66.7% (*n* = 12) for LSG and LGCP groups, respectively (*p* = 0.004). The LSG group was significantly heavier at baseline than the LGCP group (median BMI 39.3 in LSG and 33.3 in LGCP, *p* < 0.001), and the LSG group was also associated with significantly poorer glycemic control preoperatively. However, after adjusted for confounding factors, this study further demonstrated that LSG method was capable of inducing significantly greater weight loss 3 years following surgery. 

In contrast to previous studies, we performed a longer follow-up of 3 years to evaluate the long-term outcomes between LGCP and LSG. We revealed a widened gap in excessive weight loss between LSG and LGCP. Notably, the LSG group exhibited a better weight-reduction rate than the LGCP group after 6, 12, and 24 months during the 3-year follow-up postoperatively, even though both groups achieved similar positive results for weight reduction, BMI, GOT, GPT, cholesterol, triglyceride, creatinine, HbA1c, and the estimated weight reduction rate. Moreover, the LGCP group exhibited a narrower range of the weight-reduction rate and a higher reoperation rate than the LSG group. The reasons for the reoperation of 11 patients with LGCP were weight regain and unsatisfying weight loss. Still, we observed no complications in the LGCP group, whereas four patients with LSG had postoperative complications. Three of the four patients with early complications were resolved before discharge; the fourth patient had gastric outlet obstruction and was treated with Roux-en-Y surgery. No deaths occurred in either groups. Different publications have reported short- and mid-term outcomes demonstrating the effectiveness of both LSG [7,24], but only a handful of studies reported long-term outcomes for plication [48]. Abdelgawad et al. reported that LGCP was associated with poor weight loss (%EWL: 32% and weight regain: 58.3%) at the 6-year follow-up visit which resulted in a high rate of revisions [48]. 

### Limitations

Our results must be interpreted within the limitations of this study, owing to its nature as a retrospective file review. We could not present data about patient satisfaction, and the study population was relatively small. Thus, prospective randomized trials are needed to further investigate the effectiveness of these two bariatric surgery techniques.

## 5. Conclusions

Our long-term follow-up study reveals that LGCP appears inferior to LSG in achieving excessive weight loss during the first 3 years postoperatively. LGCP also presented a higher reoperation rate due to unsatisfactory weight loss and weight regain. Hence, we argue that the role of LGCP in bariatric surgery should be critically reappraised, mainly when LSG is a feasible alternative.

## Figures and Tables

**Figure 1 medicina-58-01299-f001:**
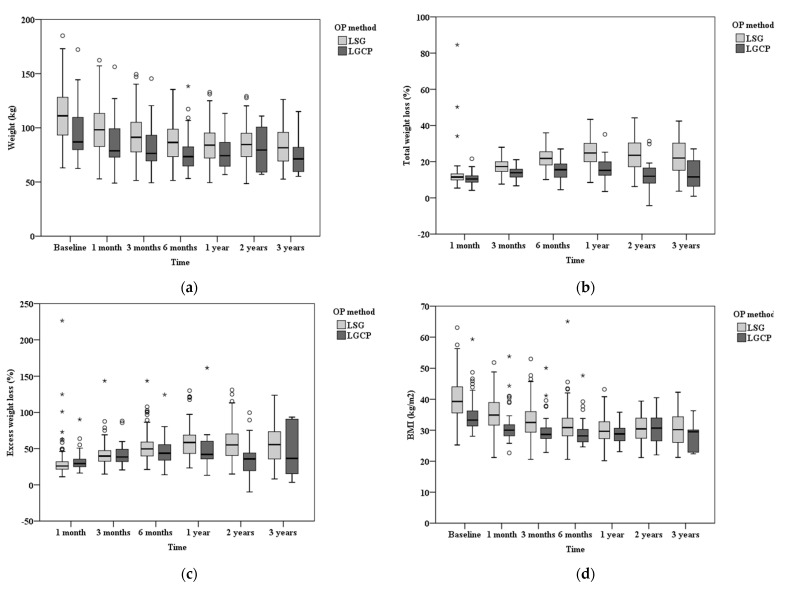
Changes in (**a**) weight, (**b**) % total weight loss, (**c**) % excess weight loss, and (**d**) body mass index (BMI) between laparoscopic greater curvature plication (LGCP) and laparoscopic sleeve gastrectomy (LSG) groups over long-term follow-up. When a value differs by more than 1.5 × IQR from the range of the first and third quartiles, the value is an outlier. Both circles and asterisks are outliers. The circles represent values outside the range of 1.5 × IQR to 3 × IQR, called mild outliers. Asterisks indicate values that are outside the range of more than 3 × IQR and are called extreme outliers.

**Table 1 medicina-58-01299-t001:** Demographic description of the patients in this study.

	OP Method				
	LSG (n = 231)		LGCP (n = 49)		
	Median	IQR	Median	IQR	*p*-Value
Age	34.0	27.0–41.0	34.0	30.0–41.0	0.768
Height	167.0	160.0–173.0	163.0	158.0–167.0	0.028
Weight	111.1	93.3–128.4	87.0	79.9–109.7	<0.001
Ideal weight	61.3	56.3–65.8	58.5	54.9–61.4	0.028
Weight excess	49.0	36.4–64.1	28.5	24.3–40.8	<0.001
BMI-baseline	39.3	35.6–44.1	33.3	31.4–36.2	<0.001
BMI-end	30.2	26.1–34.4	29.5	22.9–30.1	0.472
GOT-baseline	29.0	21.0–42.0	25.0	20.0–35.0	0.083
GOT-end	19.0	17.0–24.0	22.5	21.0–27.0	0.211
GPT-baseline	36.0	24.0–60.0	31.0	17.0–42.0	0.022
GPT-end	17.0	13.0–24.0	22.0	18.0–34.0	0.259
Cholesterol- baseline	187.0	163.0–220.0	194.0	168.0–219.0	0.751
Cholesterol-end	179.0	148.0–191.0	202.0	178.0–215.0	0.062
TG-baseline	145.0	95.0–180.0	121.0	82.0–175.0	0.103
TG-end	93.0	49.0–120.0	52.5	41.0–126.0	0.559
Uric acid-baseline	6.4	5.4–7.4	5.7	5.1–6.7	0.010
Uric acid-end	5.8	5.5–6.6	4.8	4.4–5.5	0.197
Creatinine-baseline	0.7	0.6–0.8	0.7	0.6–0.7	0.067
Creatinine-end	0.8	0.7–0.8	0.7	0.6–0.8	0.382
HbA1c-baseline	5.8	5.4–6.5	5.5	5.3–5.7	0.001
HbA1c-end	5.4	5.1–5.6	5.3	5.2–5.5	0.901
Glucose AC-baseline	96.0	89.0–111.0	91.0	87.0–102.5	0.039
Glucose AC-end	87.0	81.0–97.0	87.0	83.0–96.0	0.969
Operative time(min)	159.0	132.0–193.0	183.0	155.0–220.0	<0.001
Length of stay(day)	4.0	4.0–5.0	4.0	4.0–5.0	0.017

IQR: interquartile range. *p*-value by Mann-Whitney U Test. Note: Laparoscopic sleeve gastrectomy, LSG; Laparoscopic greater curvature plication, LGCP; body mass index, BMI; glutamic oxaloacetic transaminase, GOT; glutamic pyruvic transaminase GPT; triglyceride, TG; hemoglobin A1c (HbA1c).

**Table 2 medicina-58-01299-t002:** Demographic characteristics of the health profile of the patients in this study.

		OP Method						
		LSG (*n* = 231)		LGCP (*n* = 49)		Total (*n* = 280)		*p*-Value
Baseline		N	%	N	%	N	%	
Gender	Female	133	57.6	36	73.5	169	60.4	0.039
	Male	98	42.4	13	26.5	111	39.6	
Hypertension	No	125	54.1	30	61.2	155	55.4	0.363
	Yes	106	45.9	19	38.8	125	44.6	
Hyperglycemia	No	167	72.3	43	87.8	210	75.0	0.023
	Yes	64	27.7	6	12.2	70	25.0	
Hyperuricemia	No	216	93.5	45	91.8	261	93.2	0.753
	Yes	15	6.5	4	8.2	19	6.8	
Osteoarthritis	No	223	96.5	44	89.8	267	95.4	0.057
	Yes	8	3.5	5	10.2	13	4.6	
Fatty liver	No	216	93.5	39	79.6	255	91.1	0.005
	Yes	15	6.5	10	20.4	25	8.9	
Fatty liver severity	1	0	0.0	1	10.0	1	4.0	0.155
	2	1	6.7	3	30.0	4	16.0	
	3	1	6.7	1	10.0	2	8.0	
	4	2	13.3	2	20.0	4	16.0	
	5	6	40.0	3	30.0	9	36.0	
	6	5	33.3	0	0.0	5	20.0	
Re-operation	No	227	98.3	42	85.7	269	96.1	0.001
	Yes	4	1.7	7	14.3	11	3.9	
Complication	No	227	98.3	49	100.0	276	98.6	1.000
	Yes	4	1.7	0	0.0	4	1.4	
Acute respiratory failure	No	229	99.1	49	100.0	278	99.3	1.000
	Yes	2	0.9	0	0.0	2	0.7	
Pneumonia	No	230	99.6	49	100.0	279	99.6	1.000
	Yes	1	0.4	0	0.0	1	0.4	
Electrolytes imbalance	No	230	99.6	49	100.0	279	99.6	1.000
	Yes	1	0.4	0	0.0	1	0.4	
Pulmonary edema	No	230	99.6	49	100.0	279	99.6	1.000
	Yes	1	0.4	0	0.0	1	0.4	
Intra-abdominal abscess	No	231	100.0	49	100.0	280	100.0	
Gastric outlet obstruction	No	230	99.6	49	100.0	279	99.6	1.000
	Yes	1	0.4	0	0.0	1	0.4	

*p*-value by Chi-Square Test or Fisher’s Exact Test when appropriated. Note: Laparoscopic sleeve gastrectomy, LSG; Laparoscopic greater curvature plication, LGCP.

**Table 3 medicina-58-01299-t003:** Results of multiple generalized linear models with log link and gamma distribution in GEE method on %EWL and %TWL.

		%EWL			%TWL		
Parameter		Mean Ratio	95% CI	*p*-Value	Mean Ratio	95% CI	*p*-Value
(Intercept)		90.400	64.844–126.028	<0.001	9.600	7.502–12.285	<0.001
Age		1.000	0.996–1.003	0.961	0.999	0.996–1.002	0.559
Gender	Male	1.097	1.007–1.194	0.033	1.107	1.019–1.203	0.016
	Female	1.000			1.000		
BMI (at before surgery)		0.971	0.964–0.978	<0.001	1.006	1.000–1.011	0.042
Hyperglycemia	Yes	0.952	0.882–1.029	0.216	0.970	0.903–1.043	0.415
	No	1.000			1.000		
Fatty liver	Yes	1.043	0.907–1.199	0.555	1.043	0.924–1.177	0.499
	No	1.000			1.000		
OP method (at 1 month after surgery)	LGCP	0.943	0.833–1.069	0.359	0.922	0.826–1.030	0.152
	LSG	1.000			1.000		
Time from surgery (in LSG)	3 years	1.922	1.696–2.179	<0.001	1.894	1.661–2.158	<0.001
	2 years	1.971	1.807–2.150	<0.001	1.965	1.799–2.145	<0.001
	1 year	2.032	1.886–2.189	<0.001	2.036	1.888–2.195	<0.001
	6 months	1.784	1.668–1.909	<0.001	1.788	1.669–1.915	<0.001
	3 months	1.433	1.346–1.525	<0.001	1.436	1.346–1.531	<0.001
	1 month	1.000			1.000		
Interaction of OP method and time:							
OP method (LGCP vs. LSG)	3 years	0.675	0.450–1.014	0.058	0.671	0.451–0.999	0.049
	2 years	0.679	0.502–0.918	0.012	0.679	0.502–0.918	0.012
	1 year	0.749	0.630–0.890	0.001	0.749	0.639–0.878	<0.001
	6 months	0.801	0.701–0.914	0.001	0.804	0.708–0.912	0.001
	3 months	0.911	0.827–1.003	0.058	0.913	0.830–1.004	0.061
	1 month	1.000			1.000		

%EWL = Weight reduction (kg)/Ideal weight reduction (kg) × 100. %TWL = Weight reduction (kg)/baseline weight (kg) × 100. GEE: Generalized Estimating Equations. Note: Laparoscopic sleeve gastrectomy, LSG; Laparoscopic greater curvature plication, LGCP; body mass index, BMI.

**Table 4 medicina-58-01299-t004:** Estimated weight reduction rate (%EWL and %TWL) by treatment group for participants in the long-term follow-up.

		OP Method					
		LSG			LGCP		
	Time	Mean	SE	95% CI	Mean	SE	95% CI
%EWL	3 years	56.6	61.9	6.6–483.2	36.0	40.2	4.1–320.4
	2 years	58.0	63.8	6.7–501.2	37.2	41.1	4.3–324.6
	1 year	59.8	65.9	6.9–517.8	42.2	46.9	4.8–373.0
	6 months	52.5	57.8	6.1–453.9	39.6	44.4	4.4–356.0
	3 months	42.2	46.4	4.9–364.0	36.2	40.6	4.0–325.1
	1 month	29.4	32.3	3.4–253.7	27.8	31.1	3.1–249.5
%TWL	3 years	23.1	20.0	4.3–125.9	14.3	12.6	2.6–79.9
	2 years	24.0	20.9	4.3–132.5	15.0	13.0	2.7–82.3
	1 year	24.9	21.7	4.5–137.8	17.2	15.1	3.1–96.0
	6 months	21.8	19.1	3.9–121.0	16.2	14.4	2.8–92.0
	3 months	17.5	15.3	3.2–97.3	14.8	13.1	2.6–83.8
	1 month	12.2	10.7	2.2–67.9	11.3	10.0	2.0–64.1

Adjusted for age, gender, BMI at baseline, hyperglycemia, and fatty liver.

## Data Availability

The data presented in this study are available upon request from the corresponding author.

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
