# Peer review of "Long-Term Outcomes of Laparoscopic Greater Curvature Plication and Laparoscopic Sleeve Gastrectomy: Critical Appraisal of the Role of Gastric Plication in Bariatric Surgery"

_medicina, 2022, doi:10.3390/medicina58091299_

Round 1

Reviewer 1 Report

Thank you for opportunity to review your manuscript. You showed the Long-term outcomes of laparoscopic greater curvature plication and laparoscopic sleeve gastrectomy with critical appraisal in this manuscript.

Laparoscopic greater curvature plication was reported since 2007, and many papers including meta-analyses were reported.

This manuscript is interesting, however there are little new findings or novelty.

Reviewer 2 Report

The article deals with a very interesting issue in bariatric surgery comparing two different techniques  , Laparoscopic Greater Curvature Plication and Laparoscopic Sleeve Gastrectomy. The study is very well organized . The number of patients is appropriate. Both surgical methods are well analyzed. My only concern is that the operation time in both methods is relatively longer than the usual. The extended follow up is for benefit for a better analysis of the results of both methods.   The end results are very interesting and helpful for younger surgeons.  

Reviewer 3 Report

This is a single center, retrospective study, to compare laparoscopy gastric plication vs sleeve gastrectomy, in a non-RCT planning. Gastric plication procedures should be considerer investigational at present; so, its needs institutional review board and, preferible, a registered on a platform for clinical trials.

This study includes cases of unmatched patients, so its results are descriptive and exploratory.

Some changes are necessary to improve the results:

1.- Weight loss should be expressed as a percentage of total body weight (%TBW) and not as %EWL (higher BMIs erroneously induce lower %EWLs).

2.- Indicate for each surgical procedure:

 %TBW at nadir and at 2-3 years of follow-up.

% of subjects who have achieved a weight loss > 25-30% at nadir and 3 years of follow-up. Otherwise surgical failure (% of patients with %TBW<20% at nadir and 3 years.

% of patients with type 2 diabetes at baseline. Remission of diabetes at 3 years. In the same way, remission of other major comorbidities (HTN, OSA, Dyslipidemia).

3.- Fatty liver, how was it diagnosed? by clinical data or intraoperative biopsy?

4.- Please reduce the figures. Those related to the evolution of BMI and %TBW are sufficient

5.- For weight and biochemistry variables, a simple table (with means +/- SD and median with  interquartile range), at baseline and at the end of follow-up, is sufficient. (It is not necessary to replicate it in figures).

6.- Rewrite the discussion according to the new changes of the obtained results.

Round 2

Reviewer 1 Report

Thank you for opportunity to review your revised manuscript. You showed the Long-term outcomes of laparoscopic greater curvature plication and laparoscopic sleeve gastrectomy with critical appraisal in this manuscript.I commented that author’s manuscript is interesting, but there are little new findings or novelty. However, author adjusted for confounding factors and demonstrated new important results. Main document and Tables were reconstructed.

Reviewer 3 Report

This study describes the inferiority of the gastric plication technique vs LSG, which is a strong point for its publication, but the content of the article can be improved.

1.- %TWL vs %EWL

Despite the explanations provided by the authors, %EWL is not the best way to express weight loss. Bariatric surgery journals recommend %TWL or changes in BMI along with %EWL (which should refer to an ideal weight corresponding to BMI=25).

Examples:

Obes Surg (instructions to authors):

4b. TERMINOLOGY Please follow the mandatory manuscript terminology standards. · Weight loss must be expressed as change in BMI or %total weight loss (%TWL)

SOARD (Author information)

Weight loss must be expressed as change in BMI, AS WELL AS % Excess Weight Loss (%EWL), with the calculation of ideal body weight as that equivalent to a BMI of 25 kg/m2 and/or % Excess BMI Lost (%EBMIL) with excess BMI > 25 kg/m2 AS WELL AS % total body weight loss.

In addition, you can also review the following article:

van Rijswijk AS, van Olst N, Schats W, van der Peet DL, van de Laar AW. What Is Weight Loss After Bariatric Surgery Expressed in Percentage Total Weight Loss (%TWL)? A Systematic Review. Obes Surg. 2021 Aug;31(8):3833-3847. doi: 10.1007/s11695-021-05394-x. Epub 2021 May 17. PMID: 34002289.

So, it is a rule of thumb to express weight loss alternatively in another way (%TWL or changes in BMI). It should be incorporated into the material and methods, and it is not necessary to transfer it to the discussion.

2.- I would appreciate it if the tables were simplified with a central measure (mean or median) and dispersion (SD or IQR). It is not necessary show all the descriptive measures (mean, median, Q1,Q3, min, max).

3.- This recent paper should be commented on in the discussion, to strengthen the results of this study:

Abdelgawad M, Elgeidie A, Sorogy ME, Elrefai M, Hamed H, El-Magd EA. Long-Term Outcomes of Laparoscopic Gastric Plication for Treatment of Morbid Obesity: a Single-Center Experience. Obes Surg. 2022 Aug 12. doi: 10.1007/s11695-022-06217-3. Epub ahead of print. PMID: 35962269.

4.- TITLE: Please change for the correct word.

 Role of Gastric Placation in Bariatric Surgery

Round 3

Reviewer 3 Report

No comments